# Self-supervised Reinforcement Learning with Independently Controllable Subgoals

Andrii Zadaianchuk[1,2], Georg Martius[1], Fanny Yang[2]
[1] Max Planck Institute for Intelligent Systems, Tübingen, Germany
[2] Department of Computer Science, ETH Zurich
andrii.zadaianchuk@tuebingen.mpg.de

**Abstract:** To successfully tackle challenging manipulation tasks, autonomous agents must learn a diverse set of skills and how to combine them. Recently, self-supervised agents that set their own abstract goals by exploiting the discovered structure in the environment were shown to perform well on many different tasks. In particular, some of them were applied to learn basic manipulation skills in compositional multi-object environments. However, these methods learn skills without taking the dependencies between objects into account. Thus, the learned skills are difficult to combine in realistic environments. We propose a novel self-supervised agent that estimates relations between environment components and uses them to independently control different parts of the environment state. In addition, the estimated relations between objects can be used to decompose a complex goal into a compatible sequence of subgoals. We show that, by using this framework, an agent can efficiently and automatically learn manipulation tasks in multi-object environments with different relations between objects.

**Keywords:** object-centric representations, relations, self-supervised reinforcement learning

## 1 Introduction

Autonomous agents that need to solve manipulation tasks in environments with many objects have to master a variety of skills. In addition, such agents should be able to properly combine these skills to solve complex tasks. In modular environments, the agent must explore many different ways how it can control the environment [1]. Self-supervised agents that imagine their own goals can automate this process, and learn many skills without external reward signals [1, 2, 3, 4, 5, 6, 7]. One of the main challenges for goal-based autonomous agents is the choice of a suitable goal space and the corresponding reward function [8]. As this choice determines the difficulty of the learning, it is crucial to exploit all available structure in the environment state for construction of the goal space.

One natural way to represent the state in modular environments is to use an object-centric representation: the environment state is represented as a set of components, with each component corresponding to the state of an individual object [9, 10, 11]. Such representations can be learned in an unsupervised fashion from high-dimensional observations such as images [12, 13, 14, 9, 15, 16, 17]. Therefore, methods that use object-centric representations can be readily extended to take high-dimensional data as input. A simple approach to use object-centric representations in autonomous learning is to first learn how to control each object individually (using the objects' representations as subgoals), and then combine learned skills to control multiple objects [10]. However, in an environment where different objects interact with each other, this method might learn an *incompatible* sequence of skills, i.e. achieving one of the subgoals can destroy another previously achieved subgoal. For example, moving one object from a stack of objects may change the position of the others.

One line of work that aims at learning sequences of skills that are compatible is Hierarchical Reinforcement Learning (HRL) [18, 19, 20]. In principle, hierarchical agents should be able to transform a task into a sequence of subtasks that they solve sequentially. However, to date, existing hierarchical agents have mostly been applied to learn navigation or reaching tasks where learned skills do

5th Conference on Robot Learning (CoRL 2021), London, UK.

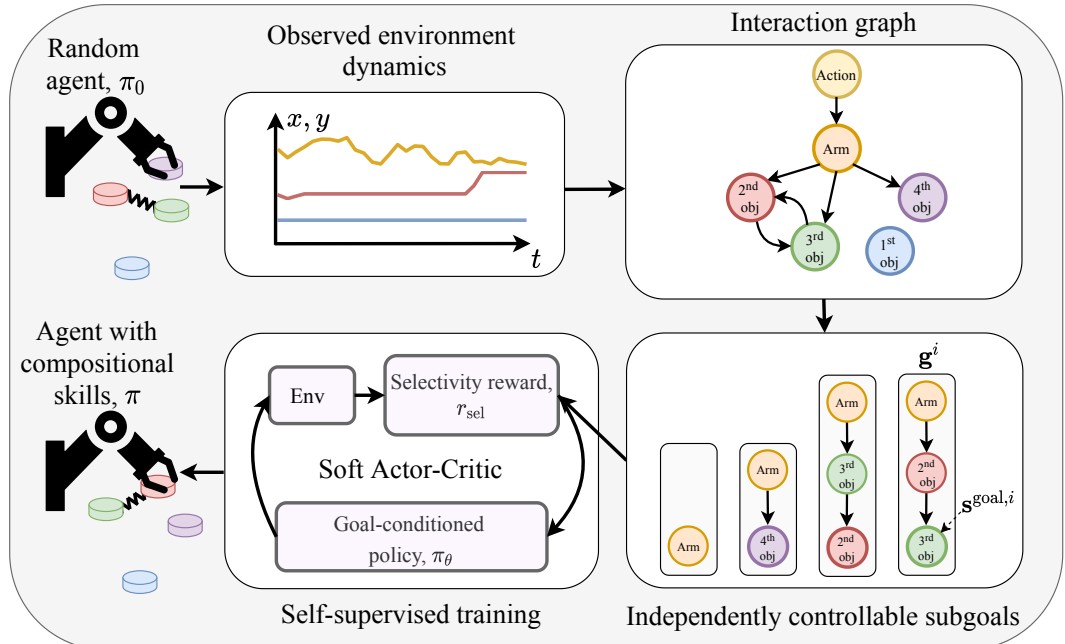

Figure 1: Our SRICS method. First, the interaction graph is inferred from observed environment dynamics containing links from cause to affected entity. This gives rise to subspaces that can be independently controlled, corresponding to subgoals $\mathbf{g}^i$. Next, the subgoals $\mathbf{g}^i$ are used to construct a selectivity reward signal $r_{\mathrm{sel}}$. The selectivity reward $r_{\mathrm{sel}}$ incentivizes the agent to only control the main entity $i$ towards $\mathbf{s}^{\mathrm{goal},i}$ within each subgoal $\mathbf{g}^i$ without affecting entities outside the subgoal. SRICS learns to solve an external goal $\mathbf{s}^{\mathrm{goal}}$ by decomposing it into an ordered list of subgoals $\mathbf{g}^i$ and solving each using SAC [24] with a goal-conditioned policy $\pi_\theta$. As a result, the agent attempts to solve all the discovered subgoals one-by-one, without destroying previously solved subgoals.

not interact with each other. It is unclear how sensitive hierarchical agents are to possible interactions between learned skills. In this paper, we investigate another approach by reformulating the agent's subtasks and the corresponding reward signals. Similar to Thomas et al. [21], we train an agent such that it is motivated to control a particular component of the environment state representation while minimally affecting other components. Such an agent can learn to control components independently from other components, thus making the learned skills compatible with each other.

As the environment state representation is not necessarily disentangled as in Thomas et al. [21], our method should additionally account for possible relations between components. We propose a novel *selectivity reward signal* that uses an *interaction graph* to determine a set of components that can be selectively controlled without interacting with the remaining scene. The interaction graph can be inferred from observed objects dynamics collected by a random policy without supervision [22, 23]. Thus, we combine learning of such interaction graphs with a goal-conditional reinforcement learning (RL) method that operates on object-centric representations [10] and uses the selectivity reward signal. During training (schematically depicted in Fig. 1), our SRICS agent (for **S**elf-supervised **R**elational RL with **I**ndependently **C**ontrollable **S**ubgoals) learns how to efficiently achieve different subgoals (and control the corresponding subspaces) while being incentivized to minimize its effects on other parts of the environment.

Our main contributions are as follows:

- We show that the global interaction graph can be estimated from data using a recurrent graph neural network (GNN) dynamical model combined with a sparsity prior.
- We propose a *goal-directed selectivity reward function* that allows an agent to learn how to control environment components independently from one another.
- We develop SRICS, an algorithm that uses the inferred interaction graph to learn simple and independently controllable subtasks and decompose a complex goal into a compatible sequence of subgoals.

## 2 Modular Goal-conditional Reinforcement Learning

We are interested in an agent that can solve multiple tasks in an environment. In particular, we consider goal-based task encodings where each task corresponds to an environment state the agent has to reach, denoted as the goal state $\mathbf{g}$. The task is then given to the agent by conditioning its policy $\pi(\mathbf{a}_t \mid \mathbf{s}_t, \mathbf{g})$ on the goal $\mathbf{g}$, and the agent's objective is to maximize the expected goal-conditional return:

$$\mathbb{E}_{\mathbf{g} \sim P} \left[ \sum_{t=1}^{T} \mathbb{E}_{\mathbf{s}_t \sim \rho_\pi, \mathbf{a}_t \sim \pi, \mathbf{s}_{t+1} \sim d} \left[ r(\mathbf{s}_{t+1}, \mathbf{g}) \right] \right]$$

where $d$ is an unknown dynamics distribution, $\rho_\pi$ is the state marginal distribution induced by the agent's policy $\pi$ and $P$ is some distribution over the space of goals $\mathcal{G}$ that the agent receives for training. A common approach to define the reward function in this setting is the negative distance of the current state to the goal: $r(\mathbf{s}, \mathbf{g}) = -\|\mathbf{s} - \mathbf{g}\|$. In general, however, the goal space $\mathcal{G}$ does not need to be equal to the state space $\mathcal{S}$, but can be any task embedding space with potentially different dimensionality [1, 25]. As some tasks cannot be expressed as desired regions of the state space, the goal $\mathbf{g}$ can parameterize a more general objective $r(\mathbf{s}, \mathbf{g})$ that the agent should maximize. Many environments are modular, in the sense that an agent's overall goal (e.g. manipulating many objects) can be decomposed into different *subgoals* (e.g. manipulating individual objects) that can be sequentially achieved.

### 2.1 Object-Centric Representations

We use object-centric representations for the state space $\mathcal{S}$. That is, the state space $\mathcal{S}$ is a direct product of all object subspaces, $\mathcal{S} = \mathcal{S}^1 \times \ldots \times \mathcal{S}^K$, where each $\mathcal{S}^j$ corresponds to the state of an entity. The state of an entity is encoded by its position $s^{j,\text{where}}$ and an identifier $s^{j,\text{what}}$. The semantics behind an entity are unknown to the agent, i.e. the state of both the agent and the objects to manipulate are encoded identically. The agent has no information about which objects are controllable and how they are related. Object-centric representations could be learned in an unsupervised way from sequential image data [9, 12, 13] and learning them is an orthogonal line of research. In this work, we focus on using object-centric representations for decomposing the goal to subgoals that are compatible with each other and can be achieved sequentially to solve the original goal.

The choice of the goal space plays a crucial role in determining the difficulty of the learning task. If the environment state consists of independent parts, it is easiest to learn to control these components independently [10]. However, in the case of interactions between these components, learning to achieve subgoals in such environments and combining learned skills could be harmful to achieving the original compositional goal. For example, assume that the subgoals consist of moving objects to different positions. By solving a single subgoal without taking the other subgoals into consideration, the agent might unintentionally rearrange objects from previous subtasks, resulting in an overall deterioration instead of an improvement. In the next section, we present a method that accounts for such dependencies by learning an interaction graph to decompose a goal into *independently controllable subgoals* and introduce a corresponding reward function.

## 3 Self-Supervised Relational RL with Independently Controllable Subgoals

In the setting we consider, at the training stage, the agent only receives a single compositional goal from the environment. The agent could try to solve the goal using the usual negative distance to the goal as a reward signal. However, achieving the compositional goal is quite a complex task by itself. This challenge can be addressed by discovering simple skills and combining them to solve the compositional goal. To achieve this, the agent needs to rely on self-supervision in the form of splitting the goal into subgoals and internally constructing the reward signal connected to each subgoal.

The agent uses data collected from the environment to discover how different parts of the environment are related, including the agent itself, and then uses the discovered relations for the construction of subtasks that are solvable and can be easily combined. First, we describe how to use object-centric representations to estimate a graph of relations between objects, and then show how to utilize the learned graph during agent training, and for goal decomposition during evaluation.

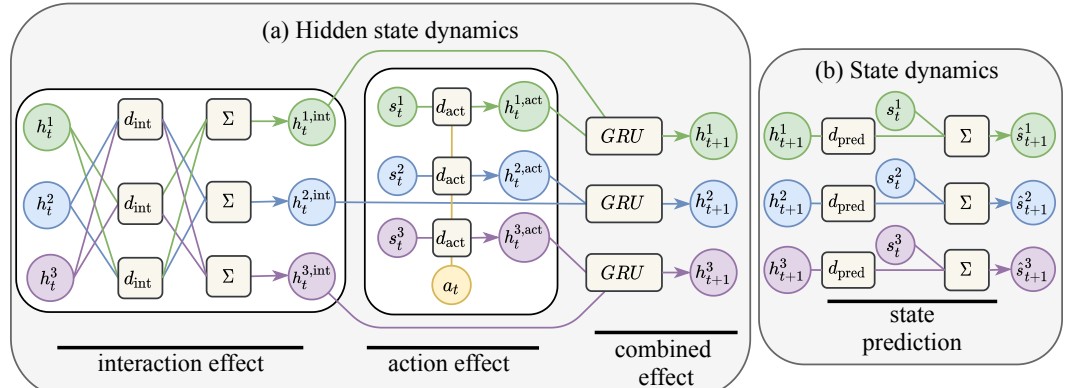

Figure 2: The dynamical model. For a given object $j$, the function $d_{\text{int}}$ computes each of the other objects' effect on the object $j$ using the hidden states $\mathbf{h}_t$. The effects from all the other objects are aggregated in the interaction effect vector $\mathbf{h}_t^{j,\text{int}}$. Next, the function $d_{\text{act}}$ computes the action's effect $\mathbf{h}_t^{j,\text{act}}$ on the object $j$. Both effects are combined in the GRU. Finally, object's state estimation $\hat{\mathbf{s}}_{t+1}^{j}$ is estimated from the hidden state $\mathbf{h}_{t+1}^{j}$ using the prediction function $d_{\text{pred}}$.

## 3.1 Estimation of the Latent Interaction Graph with a GNN Dynamical Model

Relational information in the environment can help the autonomous agent to gain control over different parts of the environment. For example, if some parts of the environment cannot be affected, the agent will be more efficient by not trying to control them. Recently, several methods to estimate this relational information in an unsupervised way were proposed [23, 26, 27, 28, 29]. Most of them assume that the relations are static [23, 28, 30]. As this is not the case in many robotic manipulation applications, we propose to use a similar approach suitable for constantly changing relations. For this, we use a graph neural network (GNN) [22, 31] to model the forward dynamics of the objects.

Because states could be non-Markovian, we use a recurrent dynamical model. Specifically, we incorporate recurrence in the GNN model by adding a Gated recurrent unit (GRU) [32] to the GNN message passing operation (see Fig. 2a). We use the functions $d_{\text{int}}$ and $d_{\text{act}}$ to model the object-object interaction effect and the action's effect, respectively. Next, both effects are combined in the GRU. More formally:

$$\mathbf{h}_t^{j,\text{int}} = \sum_{i \neq j} d_{\text{int}}\left(\mathbf{h}_t^i, \mathbf{h}_t^j\right), \quad \mathbf{h}_t^{j,\text{act}} = d_{\text{act}}\left(\mathbf{s}_t^j, \mathbf{a}_t\right), \quad \mathbf{h}_{t+1}^j = \text{GRU}\left(\left[\mathbf{h}_t^{j,\text{int}}, \mathbf{h}_t^{j,\text{act}}\right], \mathbf{h}_t^j\right), \quad (1)$$

where $\mathbf{h}_t^{j,\text{int}}$ and $\mathbf{h}_t^{j,\text{act}}$ are vectors representing interaction and action effects, whereas $\mathbf{h}_t^j$ is the hidden state for object $j$ at time step $t$.

To model dynamics with sparse interactions between objects, we model $d_{\text{int}}$ as the product of an interaction weight $w_t^{ij} \in \{0, 1\}$ and an interaction effect function $d_{\text{int-eff}}$:

$$d_{\text{int}}(\mathbf{h}_t^i, \mathbf{h}_t^j) = w_t^{ij} \cdot d_{\text{int-eff}}(\mathbf{h}_t^i, \mathbf{h}_t^j). \quad (2)$$

The interaction weight $w_t^{ij}$ represents the belief in the absence or presence of the interaction between object $i$ and object $j$ at time step $t$. We model the weight's distribution as

$$q\left(w_t^{ij} \mid \mathbf{s}_t\right) = \text{softmax}\left(d_{\text{int-pres}}(\mathbf{h}_t^i, \mathbf{h}_t^j)\right), \quad (3)$$

where $d_{\text{int-pres}}$ is the interaction presence function. As we are interested in the estimation of the connections that are necessary for predictions, we additionally encourage the interaction weights distribution $q\left(w_t^{ij} \mid \mathbf{s}_t\right)$ to be close to the sparsity prior $p_{\text{prior}}$. In our case, the sparsity prior $p_{\text{prior}}$ is the Bernoulli distribution with a large probability for zero (see App. G).

Finally, we use a function $d_{\text{pred}}$ to predict the change in coordinates (see Fig. 2b):

$$\hat{\mathbf{s}}_{t+1}^{j,\text{where}} = \mathbf{s}_t^{j,\text{where}} + d_{\text{pred}}\left(\mathbf{h}_{t+1}^j\right). \quad (4)$$

All functions in Eqs. 1–4 are modeled by small MLPs with parameters $\phi$.

Now, as we defined all the parts of the GNN dynamical model, we describe how to estimate the interaction graph using a variational approach. First, similar to [23], we train our model by minimizing the negative ELBO loss:

$$\mathcal{L}(\phi) = \sum_{j=1}^{K} \sum_{t=1}^{T-1} \frac{\left\| \mathbf{s}_{t+1}^{j,\text{where}} - \hat{\mathbf{s}}_{t+1}^{j,\text{where}} \right\|^2}{2\sigma^2} + D_{\text{KL}}(q \,\|\, p_{\text{prior}}), \tag{5}$$

where $\hat{\mathbf{s}}^{j,\text{where}}$ is the prediction of the position of object $j$, $\sigma^2$ is a fixed variance parameter and $D_{\text{KL}}$ denotes the Kullback-Leibler divergence. After training, we predict interaction weights $w_t$ for each timestep independently, then we average them across the whole dataset. Next, we estimate the global interaction graph by thresholding the average interaction weights to find the most active relations. Finally, we identify which object is directly controlled by the actions by finding the node that is most correlated with the action variable $\mathbf{a}_t$. In the graph of our running example as in Figure 1, we denote this node as "arm" since in all experiments the identified node corresponds to a simulated robot arm. We add the action node with index 0 and the corresponding edge to the most correlated object to the graph (see App. G for the details and the graph learning results).

## 3.2 Learning to Independently Control Objects using the Interaction Graph

In this section, we show how the agent can use the learned interaction graph to solve compositional goal $\mathbf{s}^{\text{goal}}$ that consists of goals for individual objects $\mathbf{s}^{\text{goal},i}$. The SRICS agent sequentially gains control over the objects without affecting the previously moved objects. To achieve this, the SRICS method first identifies a set of objects $\mathcal{P}^i$ that could be used to actively control object $i$ by analyzing the discovered relations in the interaction graph. For each node $i$, we find the set $\mathcal{P}^i$ of all nodes that lie in a path from the action node 0 to object node $i$. These ancestral nodes $\mathcal{P}^i$ are the objects that could be used by the agent to control object $i$. All the other nodes are not required and thus should not be affected during the manipulation of object $i$.

Next, we introduce the reward signal that uses $\mathcal{P}^i$ to incentivize the agent to learn to control an object without moving others (line 8 of Alg. 1). In order to achieve this, we propose to replace the original subgoal $\mathbf{s}^{\text{goal},i}$ by a novel *independently controllable subgoal* $\mathbf{g}^i$ that consists of the subgoal $\mathbf{s}^{\text{goal},i}$ and the ancestral nodes $\mathcal{P}^i$:

$$\mathbf{g}^i = \left( \mathbf{s}^{\text{goal},i}, \mathcal{P}^i \right). \tag{6}$$

In contrast to the original notion of a subgoal which only specifies a state component $\mathbf{s}^{\text{goal},i}$ that the agent should reach, an independently controllable subgoal $\mathbf{g}^i$ also includes information about which objects should not be interacted with to reach the target state component.

Interaction graph (left) and the independently controllable subgoal $\mathbf{g}^i$ for object 3 (right).

We now formulate the *goal-directed selectivity reward signal* that explicitly incentivizes the agent to leave all objects except $i$ and $\mathcal{P}^i$ untouched. As opposed to the usual reward signal, it depends on the independently controllable subgoal $\mathbf{g}^i$ and reads:

$$r_{\text{sel},i} \left( \mathbf{s}_t, \mathbf{s}_{t-1}, \mathbf{g}^i \right) = -\|\mathbf{s}_t^i - \mathbf{s}^{\text{goal},i}\| + \alpha \cdot \left( \text{sel}_i(\mathbf{s}_t, \mathbf{s}_{t-1}, \mathcal{P}^i) - 1 \right). \tag{7}$$

The first term is the usual goal-based negative distance to the goal, which is needed to learn directed control over object $i$. The second term includes the *selectivity* that we define as

$$\text{sel}_i \left( \mathbf{s}_t, \mathbf{s}_{t-1}, \mathcal{P}^i \right) = \begin{cases} \frac{\|\mathbf{s}_t^i - \mathbf{s}_{t-1}^i\|}{\sum_{j \notin \mathcal{P}^i} \|\mathbf{s}_t^j - \mathbf{s}_{t-1}^j\|}, & \text{if subgoal is not solved;} \\ 1 - \sum_{j \notin \mathcal{P}^i \cup \{i\}} \|\mathbf{s}_t^j - \mathbf{s}_{t-1}^j\|, & \text{otherwise.} \end{cases} \tag{8}$$

The selectivity $\text{sel}_i$ incentivizes the agent to maximize its *influence* [33, 34, 35] on object $i$ while having a minimal effect on objects $j \notin \mathcal{P}^i$ (corresponding to non-ancestral nodes in graph $G$) until the subgoal corresponding to the object $i$ is solved. Selectivity reaches its maximum value of 1 when the agent changes only the state of the object $i$ without affecting any objects $j \notin \mathcal{P}^i$. In App. F, we show that selectivity naturally increases during learning to control the environment and that using it as a reward signal increases efficiency and stability.

### 3.3 SRICS Policy Architecture and Training

Similar to the SMORL agent [10], we use a goal-conditioned attention policy for achieving subgoals. This kind of policy receives a set of object-centric representations as input together with the current subgoal representation. The aforementioned approach allows us to learn several different skills using only one policy. In addition, it is compatible with a different number of objects as inputs, thus allowing to use the agent in novel situations with a different number of objects. For more details on the goal-conditioned attention policy, we refer to App. E.

SRICS can be trained with any off-policy goal-conditioned RL algorithm. In particular, we use Soft-Actor Critic (SAC) [24] with Hindsight Experience Replay (HER) [25] as a method to improve sample efficiency. The training of SRICS is presented in Alg. 1.

### 3.4 Subgoal Ordering during Evaluation

After training, the agent can be applied to more complex tasks than the simple subtasks it was trained on. During the evaluation stage (Fig. 13 in App. I), SRICS encodes the compositional goal given by the environment into a set of independently controllable subgoals. Subsequently, it orders them by the depth of the corresponding nodes in the interaction graph $G$. Due to this order, subgoals that have a large number of dependencies are attempted first and subgoals that have only a few dependencies, like the robotic arm itself, are attempted as the later subgoals. The order of the independently controllable subgoals makes them compatible with each other. For example, the agent has to first rearrange all objects that need to be manipulated and then try to "solve" the arm subgoal, without destroying the already rearranged objects. More details can be found in App. I.

## 4 Related Work

In *self-supervised reinforcement learning*, self-supervision refers to the agent constructing its own goals together with the corresponding reward signal and using them to learn to solve self-proposed goals [1, 29, 36, 37, 2, 3, 6, 38, 39, 40, 41, 42, 4, 10]. Self-supervised agents can acquire a diverse set of general-purpose robotic skills. In the case of complex tasks, it is often beneficial to discover simpler subgoals and learn to solve them [20]. From this point of view, recent hierarchical RL (HRL) agents [20, 19, 43, 44, 18, 45] that try to solve external tasks by proposing several levels of internal subgoals are also self-supervised agents.

Levy et al. [20], Nachum et al. [19] and Wang et al. [43] propose to learn several goal-conditioned policies. In the HIRO agent [19], lower-level controllers are supervised with goals that are learned and proposed automatically by the higher-level controllers. In contrast, the HAC agent [20] trains each level of the hierarchy independently of the lower levels. The I²HRL agent [43] additionally allows bi-directional communication among HRL levels and influence-based exploration to make training more stable and efficient. As such agents need to discover all the structure in the environment while learning on several levels, such approaches struggle to solve complex tasks in modular environments [46]. Next, we review agents operating in environments where some structure is given.

The SMORL agent [10] exploits learned object-centric representations for gaining control over different objects in a self-supervised way and combines the learned skills for solving more complex compositional tasks. However, Zadaianchuk et al. [10] assume independence of different objects, restricting the use of the SMORL agent to settings where objects almost do not interact with each other. CURIOUS [8] and CWYC [29] exploit the modular structure of the goal space for efficient exploration in a given goal space. Colas et al. [8] use a policy that obtains the goal module identifier together with the goal value. Blaes et al. [29] also learn a relational graph between tasks. Both agents use a given modular structure for a learning curriculum [36], however, discovered subtasks are evaluated independently.

In realistic applications, autonomous agents usually do not have any well-structured representation. Nevertheless, agents can potentially infer it from data. We cover several directions that could be useful for such structure discovery. The first line of works [13, 12, 9, 14, 16] learns object-centric representations from images or videos. Such representations could be potentially used in combination with the SRICS agent. The second line [23, 27, 26, 28, 30] studies how object relations can be discovered from data. The improvements in both of these lines could lead to more general self-supervised agents that use a discovered structure for the generation of goals.

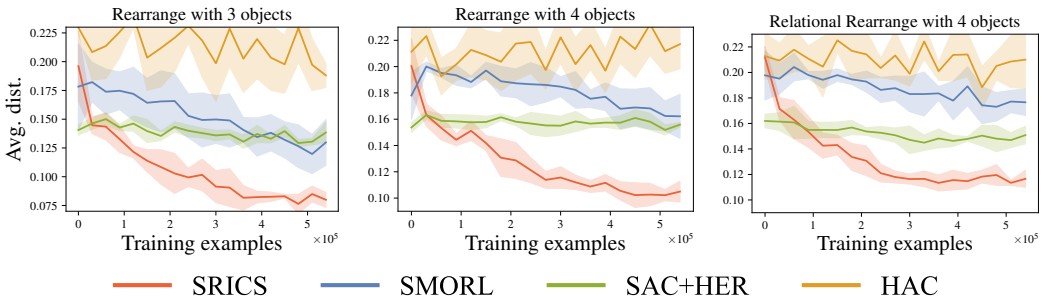

Figure 3: Average distance of objects and arm to the goal positions, comparing SRICS to SMORL, SAC+HER and HAC baselines. For all the experiments, results are averaged over 5 random seeds, shaded regions indicate one standard deviation.

## 5 Experimental Results

In this section, we present our experiments that address the following questions:

- How does SRICS perform compared to prior goal-conditioned RL methods on multi-object continuous control manipulation tasks?
- What is the performance gain obtained from the goal-directed selectivity reward and subgoal ordering during evaluation?
- How does our agent perform in an environment with an unseen combination of objects?

We run SRICS and the baseline algorithms in the *Multi-Object Rearrange* from Zadaianchuk et al. [10] and the novel *Multi-Object Relational Rearrange* environments. The latter environment incorporates additional physical connections between objects such as spring connections. Both environments are based on the `multiworld` package for continuous control tasks introduced by Nair et al. [2] and use MuJoCo [47] as a realistic simulator. They contain a 7-DoF Sawyer arm where the agent needs to manipulate a variable number of pucks on a table. In the first environment, the task is to rearrange the objects from random starting positions to random target positions. In the second environment, we add a spring connection between some of the objects and constrain other objects to be static (see App. C). This makes the resulting interaction graph more challenging and thus provides additional insights on the sensitivity of the agent to different interactions between objects. For both environments, we measure the performance of the algorithms as the average distance of all objects (including the robotic arm) to their goal positions (computed on the last step of the episode).

### 5.1 Comparative Analysis

As manipulation tasks in compositional environments can be approached from different perspectives, we provide a comparison with a state-of-the-art method from each perspective. In terms of problem assumptions, our work is closest to that of SMORL [10] which uses object-centric representations for subgoals and reward construction. In contrast to SRICS, SMORL executes subgoals in a random order and thus can potentially destroy previously solved subgoals. In addition, the SMORL agent does not have the incentive to influence the subgoal object during training. Another approach to learn goal-conditioned policy with coherent behavior is using Soft Actor-Critic (SAC) [24] with Hindsight Experience Replay (HER) [25] relabeling. This method tries to achieve the overall goal without

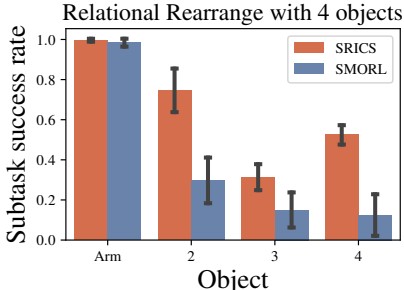

Figure 4: Subtask success rate for SRICS and SMORL for each subtask individually during evaluation in the Relational Rearrange environment. Both methods can solve Arm reaching subgoal, whereas on other subtasks SRICS performs better than SMORL.

splitting it into subgoals. Finally, we consider the Hierarchical Actor-Critic (HAC) [20] method that tries to solve compositional tasks on several levels and is state of the art on several continuous control tasks.

We show the results in Fig. 3 and Fig. 4. The performance of SRICS is significantly better than all other algorithms in both environments. SMORL is able to partially rearrange pucks on a table in the simpler *Multi-Object Rearrange* environment. However, its random subgoals ordering is inefficient for arranging all the objects including the arm. In addition, even when evaluating only based on the puck subtasks (see App. H), SRICS outperforms SMORL, which further demonstrates the benefits of using a goal-directed selectivity reward signal. Moreover, in the more complex *Multi-Object Relational Rearrange* environment, the gap between SRICS's and SMORL's performance is even larger. Furthermore, in all environments SAC is only able to solve the Arm subtasks, whereas HAC performance is close to that of a random agent. We present further comparison in more challenging environments with 6 different objects and velocity-based state representations in App. D.

## 5.2 Ablative Analysis

Here, we study the importance of different ingredients of our method for the overall performance of the agent. First, we ablate the selectivity term in our reward signal, using only the negative distance between the object and the desired position as a reward signal. We then additionally ablate the ordering of subgoals described in Sec. 3.4, using instead a random ordering of all subgoals. The results of the ablations are presented in Fig. 5. Both ablations significantly deteriorate the performance of SRICS, showing the importance of both the goal-directed selectivity reward signal and the correct ordering in the goal decomposition for object manipulation in multi-object environments.

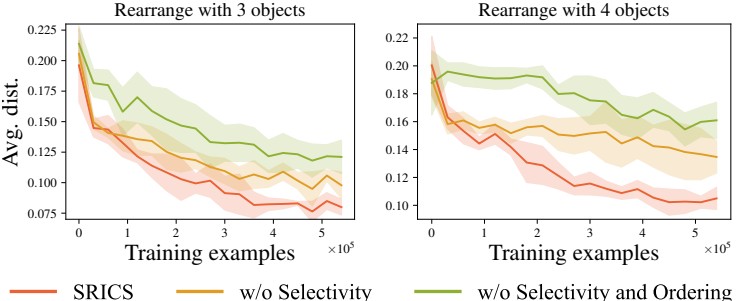

Figure 5: Average distance of objects and arm to the goal positions, comparing our method and two ablated variants on 3 and 4 objects *Rearrange* environments.

## 5.3 Generalization to Unseen Object Combinations

As SRICS can be used with different sets of objects as inputs, we investigate its performance on unseen combinations of objects. We train SRICS on the *Multi-Object Rearrange* environment, where 4 different combinations of 3 objects are presented. We leave out one of the combinations for evaluation and use the other 3 combinations for training. The performance on this modified environment is indistinguishable from SRICS's performance on *Multi-Object Rearrange* with 3 objects, showing that SRICS can operate on novel combinations of objects (details in App. B).

## 6 Conclusion and Future Work

In this work, we introduce SRICS, a self-supervised RL method that learns the relational structure of the environment and exploits this structure to learn a compatible sequence of skills to solve a difficult compositional goal. In a range of experiments in multi-object environments with robotic arm manipulation tasks, we demonstrate that SRICS is effective at discovering the most active dynamic relations between objects and can successfully rearrange multiple objects even in the presence of object interactions.

There are several interesting directions for future work. First, one can extend SRICS to image-based object-centric representations, making it more applicable to realistic robotic settings where only high-dimensional sensory information is provided as input to the agent. Moreover, we expect that SRICS can be combined with different modular curriculum learning and exploration strategies [8, 29]. Finally, we expect that active training of the dynamic interaction graph (i.e. when the data for training is collected by the agent that actively explores the environment) could further improve the discovery of important structures in the environment.

**Acknowledgments**

Andrii Zadaianchuk is supported by the Max Planck ETH Center for Learning Systems. We acknowledge the support from the German Federal Ministry of Education and Research (BMBF) through the Tübingen AI Center (FKZ: 01IS18039B).

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
