# OpenReview forum: "Self-supervised Reinforcement Learning with Independently Controllable Subgoals"
_robot-learning.org/CoRL/2021/Conference — CoRL2021 Poster_

### Official Review · Reviewer_h8jP · 2021-07-21

**Originality:** Good
**Technical Quality:** Good
**Clarity Of Presentation:** Good
**Impact:** 3

**Recommendation:**

Weak Accept: I recommend accepting the paper, but will not argue for my recommendation if the majority of other reviewers have a different opinion.

**Summary:**

The paper presents an approach that can utilize object-centric representations for solving complex goals by decomposing them into sub-goals. In particular, first the dynamics of an environment are captured in an interaction graph between different objects (using their object-centric representation such as their identity and positions). After that, various sub-graphs of the interaction graph are considered to be sub-goals. The sub-goals are ordered by the depth of their sub-graph to ensure that objects that have a lot of dependencies are attempted first, such that the corresponding sub-goals are not destroyed while attempting other sub-goals in the future. In addition, the robot is incentivized to only manipulate objects within the current sub-goal through a selectivity reward. Recurrent Graphical Neural Networks (GNN) with GRUs are used to represent the interaction graph. The goal-conditioned policy used to reach the sub-goals is trained using SAC. The method is evaluated on a simulated multi-object environment with up to 4 objects and some objects having relations between each other. It is shown to outperform a previous method SMORL, which also uses object-centric representations, a goal-conditioned baseline based on SAC with hindsight experience replay (HER), and a hierarchical actor-critic baseline.

**Issues:**

Please refer to "Strengths And Weaknesses" for a detailed list of issues.

**Reviewer Expertise:**

Good: General knowledge of the area

**Strengths And Weaknesses:**

Strengths:
- Decomposing difficult problems into sub-goals is important for scaling up robot learning. Using object-centric representations for representing sub-goals enables a significant reduction of the problem dimensionality.
- Encouraging interaction of objects only within sub-goals and ordering sub-goals by the depth of their dependency graph is important for a structured exploration without undoing previously achieved sub-goals.
- The paper is well-written and easy to understand and follow.
- The method is shown to outperform baselines in a simulated multi-object environment.

Weaknesses:
- Right now experimental evaluation only includes manipulation of a small number of simple objects. It would be interesting to see more experiments on more complicated objects and higher-dimensional environments.
- Presented method requires collecting robot interactions in order to pre-train the object interaction graph. It would be useful to know the amount of data used for pre-training to better compare it to other methods, as other methods such as SAC with HER do not require pre-training.
- The objects in the experiments are represented by their identity and position in the environment. It would be interesting to see how the method works with more examples of object representations, for example the ones that include velocity for more dynamic tasks.

**Summary Of Recommendation:**

The presented method is a great step towards a better structured learning of sub-goals. However, the scope of experiments would need to be extended in order to better understand the impact of the method.

---

> ### Author Response · Authors · 2021-08-26
> **Additional experimental results for more challenging environments and velocity-based state representations**
>
> Dear reviewer, thank you for your review and useful suggestions. Below, we address specific questions and concerns in more detail.
>
> *” Right now experimental evaluation only includes manipulation of a small number of simple objects. It would be interesting to see more experiments on more complicated objects and higher-dimensional environments.”*
>
> The choice of the environment and number of objects was done to make the experiments more comparable with prior work (SMORL). However, we agree that it is important to show how our agent performs in the case when objects have different shapes and dynamics. For these, we have implemented an additional experiment with 6 objects. Used objects are of different shapes (box and cylinder), different sizes, and different masses.
> We included the results for this environment in App. D of the revised paper.  The SRICS agent outperforms SMORL and SAC agents similar to the environment with 4 objects. As we didn’t have the resources for hyperparameter optimization during rebuttal, we used the same parameters as in the other experiments.
>
> *“Presented method requires collecting robot interactions in order to pre-train the object interaction graph. It would be useful to know the amount of data used for pre-training to better compare it to other methods, as other methods such as SAC with HER do not require pre-training.”*
>
> As mentioned in Table 2 in App. I, during pretraining, we used 1000 episodes of length 50 of random interactions, so the total amount of pretraining examples is equal to 50000. This is less than 1/10 of the data used afterward for the training (for all methods). As these are fully random actions this data could be collected once and then stored in the dataset, used for training all interaction graphs in the environment.
>
> *“The objects in the experiments are represented by their identity and position in the environment. It would be interesting to see how the method works with more examples of object representations, for example, the ones that include velocity for more dynamic tasks.”*
>
> We have picked position representation because it was shown that it is possible to estimate it from high-dimensional image data using object-centric representation learning. However, we agree that it is potentially interesting to see the difference in performance given different representations. For this, we have done additional experiments with representation that includes both position and the velocity of objects and the robotic arm. The results are presented in App. D.

---

> > ### Comment · Reviewer_h8jP · 2021-09-03
> > **Thanks**
> >
> > Thanks for addressing my comments. I have increased my score.

---

### Official Review · Reviewer_uiwp · 2021-07-23

**Originality:** Good
**Technical Quality:** Good
**Clarity Of Presentation:** Very Good
**Impact:** 4

**Recommendation:**

Weak Accept: I recommend accepting the paper, but will not argue for my recommendation if the majority of other reviewers have a different opinion.

**Summary:**

This paper presents an object-centric self-supervised reinforcement learning (RL) method in multi-goal RL settings. The major problem in prior works is that they assume each object in the environment is independently controllable without interference, which prevents the skills from being combined to solve compositional goals in realistic environments. To lift this limitation, the authors propose to learn the interaction graph between objects by recurrent graph neural network (GNN). Based on this graph, they derive a sequence of subgoals that can be controlled independently without interference, for the agent to execute. Also, they add selectivity rewards that encourage the agent to maximize the influence on the goal object while minimizing the interference to the other objects. The experimental results show that the proposed method outperforms several baselines including the recent object-centric RL and typical RL methods in multi-goal RL settings. Furthermore, the paper shows that the proposed method can generalize to unseen compositional goals.

**Issues:**

- How is selectivity compared with empowerment rewards?
Forrester, Russ. "Empowerment: Rejuvenating a potent idea." Academy of Management Perspectives 14.3 (2000): 67-80.

- In Equation 2, what's the purpose of $p(. | s_t, a_t) = \mathcal{N}(s^{j, where}_{t+1}, \sigma^2I)$? I didn't see it in other parts.

- In Equation 5, $s^{j, where}_{t}$ should be $s^{j, where}_{t+1}$?

- What is the goal space in all experiments? Is it a joint space of all objects? I suppose the baselines should use the joint space; otherwise, they cannot achieve the compositional goal. What I mean by joint space is a vector that concatenates all objects' information (e.g., positions) to a 1-D vector. I ask this question because the proposed method seems to be able to use single object information as the goal representation because the agent just needs to know a subgoal that corresponds to a single object. For a fair comparison and comprehensive study, one should use the same goal space or test both goal spaces.

**Reviewer Expertise:**

Good: General knowledge of the area

**Strengths And Weaknesses:**

Pros:
- The insight in this paper is important. Typical multi-goal RL and hierarchical RL approaches overlook the potential compositional structure and interference between objects, which limits the generalization and data efficiency. The learned interaction graph is useful for decomposing a compositional goal into several pieces of independently controllable subgoals for expediting the learning process.
- The selectivity reward is an interesting design from the self-supervised RL point of view.
Cons:
- Ordering the subgoals by the interaction graph is a sound approach to derive a sequence of independent subgoals.

Cons:
- The major issue of this paper is notation. There are too many notations in this paper without a systematically or smooth introduction to them. For example, in Section 3.1, there are many similar notations (e.g., d_oo, d_o, d_pred, d_rel-eff), but it's hard to see their relation unless you read this section multiple times. I would suggest either having a list of notations with brief explanation or plotting a figure to depict their relationship. I believe that the readers can digest these materials better.

- Lack of intuitive explanation or high-level overview before explaining the details. For example, I have no idea why do you need to estimate set $P^i$ in all paths from the action variable to object $i$? What is action variable to object $i$? Besides, I'm confused about the difference between $g^i$ and $(s^{goal,i}, P^i)$. I agree $s^{goal,i}$ is a subgoal, but why do we need $P^i$ for defining the "independently controllable subgoal"?

- Lack of baseline explanation. As SMORL is an important baseline, I would like to know more about how is SMORL different from the proposed method? For now, the paper just said SMORL uses object-centric representations and reward constructions.


**Summary Of Recommendation:**

The proposed idea is interesting and technically sound. Though the clarity and readability need lots of works, I think this idea is worth presenting at the conference. I would adjust my ratings if the author can improve their writing.

---

Post rebuttal:

The author's response addressed my questions. I've raised the score.

---

> ### Author Response · Authors · 2021-08-26
> **Significantly updated the paper with clarifications and improved notation**
>
> Dear reviewer, thank you for your positive evaluation of our work and for the constructive feedback. We have significantly updated the paper with clarifications and improved notation and hope that this addresses your concerns. We would appreciate if you could have a look in particular at Sections 3.1.-3.2. of the revised submission and l would be grateful if you could let us know in case some sections are still unclear to you. Below, we answer particular questions and specify what changes we made to the paper.
>
> *"The major issue of this paper is notation. There are too many notations in this paper without a systematically or smooth introduction to them."*
>
> Thanks for raising this concern. Based on your comment, we have fundamentally revisited our presentation for the purpose of a clearer introduction of the model. In the updated version of the paper, we introduce the (slightly modified) notation in Section 3.1 and, for the sake of an improved conceptual understanding of the dynamic model in Equations 1-4, added Figure 2. In particular, it illustrates how the different $d_{xx}$ come into play.  We modified the indexing of the functions $d_{xx}$ such that they are more intuitively aligned with their effects. Further, in the text, we now explicitly describe in words what each of the symbols such as $d_{xx}$, $h_x^y$ etc. represents. Finally, we changed the presentation order such that it is easier to follow how different parts of the model are connected.
>
> *"Lack of intuitive explanation or high-level overview before explaining the details.  For example, I have no idea why do you need to estimate set $P_i$ in all paths from the action variable to object $i$? What is the action variable to object $i$? Besides, I'm confused about the difference between $g_i$ and $(s^{\text{goal}, i}, P_i) $. I agree $s^{\text{goal}, i}$ is a subgoal, but why do we need $P_i$ for defining the "independently controllable subgoal"? "*
>
> Thanks for the question from which we learned which sections were still unclear. Section 3.2. now contains a more conceptual and intuitive introduction to the details of the selectivity reward consisting/parameterized by independently controllable subgoals. In particular, we now explicitly motivate and explain the difference of the independently controllable subgoals  $g_{i}$ compared to the original subgoals $s^{\text{goal}, i}$ and how to extract them from the interaction graph.
> Then we explain how the set of ancestral nodes $P^i$ that are all nodes in paths from the action node (now explicitly defined at the end of Section 3.1.) to the object $i$ can be used to incentivize the agent (via the selectivity reward) to reach a subgoal without interfering with previous subgoals.
>
> *"Lack of baseline explanation. As SMORL is an important baseline, I would like to know more about how is SMORL different from the proposed method?"*
>
> In addition to the discussion of  SMORL in the related work, we added the main differences between these two methods to the baselines discussion. SMORL is using distance-to-the-goal reward signal during training in contrast to the goal-directed selectivity reward signal that SRICS uses. In addition, SMORL uses random ordering of the subgoals and is hence more prone to the danger of destroying previously solved goals.
>
> *"What's the purpose of the second part of Equation 2?"*
>
> We use this distribution in the first part of the ELBO loss (the $\sigma^2$ there is the fixed variance parameter). In the new version, we introduce $\sigma$ in the loss (Eq. 5) and add a clarification directly afterward.
>
> *"In Equation 5,  $s_t^{j, \text{where}}$ should be $s_{t+1}^{j, \text{where}}$?"*
>
> We reformulate the sum indexing such that $t+1$ is used in the Eq. 5.
>
> Note that  \hat{s}_{t+1}^{j, where} is the reconstruction of the state s _{t+1}^{j, where}.
>
> *"What is the goal space in all experiments? Is it a joint space of all objects?"*
>
> We assume that the goal space is the joint space of all objects. The baselines (SAC and HAC) use it in 1-D vector representation, whereas the SMORL baseline and our SRICS method are using it part-by-part (one subgoal space is indeed only the space of one object position). After the internal iterative subgoal-achievement by SMORL/SRICS their performance is compared on the same global goal space (as SAC and HAC). In terms of execution time, all methods have the same total time to solve the compositional goal, which is split into smaller chunks for the hierarchical methods.
>
>
> *"How is selectivity compared with empowerment rewards?"*
>
> The selectivity reward tries to maximize the agent's influence on state components (as discussed in the paper). In contrast, empowerment rewards are maximizing *potential influence* to the whole state, independently of any goal. Our selectivity reward is specific to the current subgoal and also incentivizes the agent to avoid influencing other components.

---

> > ### Comment · Reviewer_uiwp · 2021-09-04
> > **Follow-up**
> >
> > Thanks for the clarification! The response addressed all of my concerns. Also, I appreciate Figure 2. I will raise the score accordingly.

---

### Official Review · Reviewer_BPut · 2021-08-03

**Originality:** Good
**Technical Quality:** Good
**Clarity Of Presentation:** Good
**Impact:** 3

**Recommendation:**

Weak Accept: I recommend accepting the paper, but will not argue for my recommendation if the majority of other reviewers have a different opinion.

**Summary:**

In this paper, a new representation learning that considers the relations between environmental objects is proposed, then a relation graph of objects is used to estimate the dependencies. Finally, the agent selects subgoals and learns to achieve them and also minimizes the influence on other objects. Experiments on locomotion tasks show the proposed method achieves higher performance compared with several previous HRL methods and representation learning methods.

**Issues:**

See the weaknesses.

**Reviewer Expertise:**

Very good: Comprehensive knowledge of the area

**Strengths And Weaknesses:**

Strengths:

The relations of objects have been investigated in MBRL, object-oriented RL, but from the perspective of HRL, it seems to have few works currently, so this paper is interesting and novel to some extent.

Experiments show some promising results.

Weaknesses:

The authors claim previous works' strong assumption of fully independent objects, while the experiments do not well support this claim.

The literature review covers many related works, but some recent works are missing, such as [1-3], where [1,2] performs better than baselines in this work.

Some formulations are not explained well. Such that the function d_{rel-eff} and d_{rel-press}, what objective ensures these functions have the effect?

I am not sure whether this kind of interaction graph restricts the scenarios to locomotion tasks. Since many atari games, some objects are not task-irrelated, so learning a subgoal as well as minimizing the effects on other objects seems not that important.

1. I2HRL: Interactive Influence-based Hierarchical Reinforcement Learning.
2. Learning Subgoal Representations with Slow Dynamics.
3. DisTop: Discovering a Topological representation to learn diverse and rewarding skills

**Summary Of Recommendation:**

Based on the review above, I give a weak accept to this paper, I think some improvements should be further made, such as the clarity of some formulations, and more analysis to support the claim and motivation, more recent baseline comparisons.

---

> ### Author Response · Authors · 2021-08-26
> **Significantly updated the paper with clarifications and the related work update.**
>
> Dear reviewer, thank you for your review. We have significantly updated the paper with clarifications and improved notation and hope that this addresses your concerns about the clarity of the formulations and the related work.
> Below, we address specific questions and concerns and clarify the changes that we made to the paper.
>
> *“The authors claim previous works' strong assumption of fully independent objects, while the experiments do not well support this claim.”*
>
>
> The baseline method SMORL is assuming independence of objects, which was also discussed in their paper in the future work section. Our experiments show that this assumption is hurting SMORL in the scenario where there are interactions between objects (e.g. in the *Relational Rearrange environment*). In Fig 3, the difference between SMORL and SRICS in the *Rearrange* tasks are to a large extent due to the modeling of the dependencies between arm and object in SRICS. This can also be seen in the ablations (Fig 5) in the difference between the yellow and the green line. Note that removing the ordering is exactly not using the information of the dependency. Additionally, when looking at the *Relational Rearrange* task, we see that SMORL is getting worse as additional dependencies between the objects are introduced.  We have edited the sentence where we discuss the “assumption of fully independent objects”, and make it more clear that other methods are not explicitly exploiting dependencies between objects.
>
> *“The literature review covers many related works, but some recent works are missing, such as [1-3], where [1,2] performs better than baselines in this work.”*
>
> Thank you for the useful pointers. We have broadened our discussion of the HRL method including these references that we were not aware of. The last reference was only recently appearing on Arxiv, so in case this work has already been published somewhere in the meantime we would be happy to update the reference.
>
>
> *”Adding more recent comparisons”*
>
> While we agree that a comparison with more promising baselines would be very useful. We want to argue that in our submission we already used well-established and some of the strongest baseline methods that could successfully tackle compositional tasks in particular. For example, as a representative from the hierarchical RL perspective, we picked HAC since it has been shown to work well for many tasks. However, for Multi-object Rearrange environments, the HAC performance is close to a random policy in the. We would be interested in hearing why the reviewer thinks the proposed methods that require similar [1] or more restrictive assumptions [2] than HAC would have a significant advantage over HAC in these environments. However, we would be very glad if you could point us to HRL methods that have concrete supporting evidence why they could outperform HAC on our compositional tasks of interest. On another note, we do think that it is an interesting research direction to find general HRL methods that can perform well on these tasks.
>
> *“Some formulations are not explained well. Such that the function $d_{rel-eff}$ and $d_{rel-press}$, what objective ensures these functions have the effect?”*
>
> Thank you for the comment based on which we have significantly updated the paper with additional clarifications (including an illustration for the different $d_{xx}$ functions) and a better introduction to the notation. To answer your question regarding $d_{rel-eff}$ and $d_{rel-pres}$: $d_{rel-eff}$ captures the effect size in the case when the effect is present, which is in turn governed by the interaction presence function $d_{rel-pres}$. $d_{rel-pres}$ is explicitly trained such that the resulting distribution after the softmax is close to a prior distribution that encourages sparsity. These aspects are now expressed more clearly as we reworded Sections 3.1 and 3.2 and added a new figure explaining the architecture of the graph network.

---

> > ### Comment · Reviewer_BPut · 2021-09-02
> > **Thanks for author's reponses**
> >
> > I think some of my concerns have been resolved with the author's response and the updated version of the paper. However, the notations should be explained in the updated version in more detail, such as the definition of two functions d_{rel-eff} and d_{rel_press}.
> >
> > As for the two previous works[1,2], I don't agree with the authors' opinion. These methods both use representation learning to solve or alleviate the non-stationary problem in HRL, which is very important in HRL literature, and I believe these provide advantages and should be compared with the proposed method.

---

### Author Response · Authors · 2021-08-27
**General Answer**

We thank all of the reviewers for their constructive feedback. Based on it, we have updated the paper with additional experimental evaluation, clarifications, and improved notation.  Below, we clarify the changes that we made to the paper to address reviewers’ concerns.

- We have conducted several additional experiments, where we show that the SRICS agent scales to the more challenging environment with 6 objects that have different shapes, sizes, and mass whereas other baselines are failing to show progress in this environment. In addition, we showed that the SRICS agent is not restricted to position-based state representation and can be used with representation that also includes the object’s velocity.
- We have fundamentally revisited our presentation for the purpose of a clearer introduction of the model in Section 3.1. There we improved the notation and added Figure 2. This figure shows the details of the dynamical model and we provide a general overview of the notation used in Section  3.1. In addition, we add several clarifications to the definition of the interactions weights and reordered parts of Section 3.1, such that it is easier to follow how different parts of the model are connected.
- We have added additional high-level explanations in Section 3.2 making it more accessible. In addition, we added the clarification of the difference between independently controllable subgoals  $g_{i}$  and usual reaching subgoals $s^{\text{goal}, i}$.
- We have edited our related work adding some missing references for the HRL part.
- We have added a better comparison between SRICS and SMORL in the discussion of the baselines.

---

### Meta-Review · Area_Chair_Hsum · 2021-08-13

**Recommendation:** Accept (Poster)
**Confidence:** 5

**Metareview:**

The reviewers are generally positive about the idea and contribution, and agree that it is important to scaling up robot learning. Post-rebuttal, they are all weakly positive.

---

### Decision · Program_Chairs · 2021-09-13

**Decision:**

Accept (Poster)

**Comment:**

The reviewers are generally positive about the idea and contribution, and agree that it is important to scaling up robot learning. Post-rebuttal, they are all weakly positive.